# Protein Kinase A (PRKA) Activity Is Regulated by the Proteasome at the Onset of Human Sperm Capacitation

**DOI:** 10.3390/cells10123501

**Published:** 2021-12-11

**Authors:** Héctor Zapata-Carmona, Lina Barón, Milene Kong, Patricio Morales

**Affiliations:** 1Laboratorio de Biología de la Reproducción, Facultad de Ciencias de la Salud, Departamento Biomédico, Universidad de Antofagasta, Antofagasta 1240000, Chile; hector.zapata@uantof.cl (H.Z.-C.); lina.baron@uantof.cl (L.B.); milene.kong@uantof.cl (M.K.); 2Instituto Antofagasta, Universidad de Antofagasta, Antofagasta 1240000, Chile

**Keywords:** sperm, capacitation, proteasome, protein kinase A, PRKA regulatory subunits, AKAP3

## Abstract

The proteasome increases its activity at the onset of sperm capacitation due to the action of the SACY/PRKACA pathway; this increase is required for capacitation to progress. PRKA activity also increases and remains high during capacitation. However, intracellular levels of cAMP decrease in this process. Our goal was to evaluate the role of the proteasome in regulating PRKA activity once capacitation has started. Viable human sperm were incubated in the presence and absence of epoxomicin or with 0.1% DMSO. The activity of PRKA; the phosphorylation pattern of PRKA substrates (pPRKAs); and the expression of PRKAR1, PRKAR2, and AKAP3 were evaluated by Western blot. The localization of pPRKAs, PRKAR1, PRKAR2, and AKAP3 was evaluated by immunofluorescence. Treatment with epoxomicin changed the localization and phosphorylation pattern and decreased the percentage of pPRKAs-positive sperm. PRKA activity significantly increased at 1 min of capacitation and remained high throughout the incubation. However, epoxomicin treatment significantly decreased PRKA activity after 30 min. In addition, PRKAR1 and AKAP3 were degraded by the proteasome but with a different temporal kinetic. Our results suggest that PRKAR1 is the target of PRKA regulation by the proteasome.

## 1. Introduction

The ubiquitin–proteasome system (UPS) is a major pathway for intracellular protein degradation [1,2]. In this system, substrates are typically marked for degradation by covalent linkages to multiple ubiquitin molecules. Ubiquitin is a monomeric protein of 76 amino acid residues and has a molecular weight of 8.5 kDa [3,4]; it is an essential element of the UPS [5]. Once marked by polyubiquitin chains, proteins are rapidly degraded by the proteasome. Proteasome function is crucial for cellular viability [6,7]. The core of the proteasome, also called 20S proteasome, is formed by two pairs of homologous rings, each containing seven subunits. The two outer rings contain α-type subunits (PSMA1-7), the function of which is to operate as a “gate” through which proteins enter the catalytic sites [8]. The β-subunits (PSMB1-7) form the two inner rings. Three of the β-subunits, β1 (PSMB6), β2 (PSMB7), and β5 (PSMB5), are catalytically active and possess caspase-like (or peptidylglutamyl-peptide), trypsin-like, and chymotrypsin-like activities, respectively [2,9,10]. Proteasomes are not static complexes, and the activity of the 20S proteasome is modulated by the binding of different regulatory complexes: 19S, PA28α/β, PA28γ, and PA200 [8,11,12]. 

Proteasomes are present in the sperm of numerous species, and several studies have shown that human sperm have all the machinery of the UPS [13,14,15,16]. More recent studies have provided compelling evidence of UPS being a regulating element of sperm capacitation [16,17,18]. Sperm capacitation was first independently reported over seven decades ago by Austin [19] and Chang [20]. This phenomenon is a process that occurs in the oviduct of the female reproductive tract, but it can be mimicked in vitro using chemically defined media. These media are based on the concentration of electrolytes and macromolecules present in the oviductal fluid. A recent study showed that oviductal fluid can modulate capacitation-associated events in vitro [21]. During the process of capacitation there are cholesterol efflux from the sperm plasma membrane and increased influx of bicarbonate and Ca^2+^, which activate soluble adenyl cyclase (SACY) to produce cAMP, leading to protein kinase A (PRKA) activation, protein tyrosine phosphorylation, actin polymerization, and the development of hyperactivated motility [22]. PRKA is a tetrameric serine/threonine (Ser/Thr) kinase holoenzyme composed of two regulatory (R) and two catalytic subunits (C). Mice sperm that lack the unique PRKA catalytic subunit Cα2 are infertile, despite normal mating behavior, and do not show the increase in tyrosine phosphorylation after sperm incubation in a capacitating medium [23], suggesting the involvement of PRKA in sperm capacitation. PRKA activity depends on the binding of four cAMP molecules to the regulatory subunits (two molecules per subunit) that allows tetramer dissociation and activation of the catalytic subunits [23]. Once free and active, the catalytic subunits phosphorylate a wide variety of substrates in the Arg-X-X-Ser/Thr motif [24,25]. 

In human spermatozoa, reports indicate that the increase in cAMP levels and activation of PRKA are among the first intracellular events of capacitation since cells incubated in a capacitating medium show a rapid (≤1 min) increase in PRKA activity, and this activity remains high and constant during this process [26,27]. This activation of PRKA leads to increased Ser/Thr protein phosphorylation, which will ultimately lead to sperm capacitation. However, intracellular levels of cAMP begin to decrease significantly after 1 min, reaching values similar to those observed in spermatozoa incubated in non-capacitating conditions at 60 min [28]. We recently described that one of the substrates of PRKACA during early stages of human sperm capacitation is the proteasome [16]. We showed that proteasome activity is directly regulated by the SACY/cAMP/PRKACA pathway. The chymotrypsin-like activity of the sperm proteasome significantly increased after 5 min of capacitation and remained high during 60 min incubation. Treatment with PRKA inhibitors significantly decreased proteasome activity. PRKA is located in the same subcellular compartments as the proteasome and co-immunoprecipitates with the proteasome. Furthermore, treatment with the proteasome-specific inhibitor epoxomicin significantly blocked capacitation and decreased the increase in tyrosine phosphorylation during capacitation. These data suggests that there might be a feedback regulation between PRKA and the proteasome, where at early incubation times PRKA phosphorylates and activates the proteasome, while at later times the proteasome regulates PRKA activity. Therefore, the aim of the present work was to evaluate the role of the sperm proteasome in regulating PRKA activity once human sperm capacitation has started. 

## 2. Materials and Methods

### 2.1. Ethics Statement 

The research presented in this manuscript was approved by the Ethics Committee on Scientific Research of the University of Antofagasta (CEIC-UA). The Ethics Committee approved the use of all human semen samples described in this study, and we followed the current guidelines for human semen studies [29]. All donors, anonymous to the researchers, signed a consent form for the use of their spermatozoa for research purposes.

### 2.2. Chemicals and Reagents 

The following reagents were purchased from Sigma Chemical Co. (St. Louis, MO, USA): Nα-tosyl-L-lysine chloromethyl ketone hydrochloride (TLCK); bovine serum albumin (BSA; A7030); Ponceau Red; HEPES; ethylenediaminetetraacetic acid (EDTA); sodium chloride (NaCl); potassium chloride (KCl); sodium phosphate dibasic (Na_2_HPO_4_); potassium phosphate monobasic (KH_2_PO_4_); calcium chloride (CaCl_2_); magnesium chloride (MgCl_2_); magnesium sulfate (MgSO_4_); sodium bicarbonate (NaHCO_3_); tris hydrochloride (Tris-HCl); Tween 20; sodium dodecyl sulfate (SDS); β-mercaptoethanol; glycerol; ammonium persulfate; N′-Tetramethylethylenediamine (TEMED); Percoll; Hoechst 33258 (H258); 1,4-diazabicyclo [2.2.2.] octane (DABCO); dimethyl sulfoxide (DMSO); sodium orthovanadate (Na_3_VO_4_); sodium fluoride (NaF); phenylmethylsulfonyl fluoride (PMSF); aprotinin; and leupeptin. The following compounds were purchased from Enzo Life Sciences (Farmingdale, NY, USA): *N*-[2-(p bromocinnamylamino) ethyl]-5-isoquinolinesulfonamide 2HCl (H89) and *N*-Acetyl-*N*-methyl-l-isoleucyl-l-isoleucyl-*N*-[(1S)-3-methyl-1-[[(2R)-2 methyloxiranyl]carbonyl]butyl]-l-threoninamide (epoxomicin). Dako Fluorescent Mounting Medium was purchased from Dako North America. 

The deionized water used in these experiments was purified to 18 megohms with an EASY-pure UV/UF ion-exchange system (Barnstead/Thermolyne, Dubuque, IA, USA). All other chemicals were of analytical grade and obtained from standard sources.

### 2.3. Culture Media

The basic medium used for all the experiments was a modified Tyrode’s medium lacking BSA and NaHCO_3_, as described [16]. This medium was designated non-capacitating medium (NCM). The capacitating medium (CM) was similar to NCM, except that it was supplemented with 2.6% BSA and 25 mM NaHCO_3_ [16]. The pH of all media was adjusted to 7.4–7.45 before use.

### 2.4. Semen Collection 

A total of 27 semen samples were provided by healthy donors aged between 20 and 30 years; some donors provided more than one sample. Freshly ejaculated spermatozoa were obtained by masturbation after 2–3 days of sexual abstinence. Samples obtained from the same donor were requested with at least a 2-week interval. Semen was collected in a sterile vessel, and each lot was tested for sperm toxicity. All semen samples used were normospermic according to World Health Organization criteria [30]. Semen samples were allowed to liquefy for 30–60 min at 37 °C. All semen samples were processed by the same person using compatible equipment, and analyses of volume, pH, sperm concentration, and percentages of motile and viable spermatozoa were performed. The mean values for semen parameters are summarized in Appendix A.

### 2.5. Sperm Selection 

Motile sperm were obtained using a dual Percoll gradient (40/80%), as described previously [15]. The final sperm pellet was resuspended in the appropriate medium at the required concentration. Approximately 5 × 10^6^ spermatozoa/mL were incubated for 0 (NCM), 1, 10, 15, 30, and 60 min (CM) at 37 °C and 5% CO_2_ in air and in the presence or absence of inhibitors. None of the inhibitors or solvents used in this study negatively affected sperm viability or motility (data not shown).

### 2.6. SDS-PAGE and Immunoblotting 

After capacitation, sperm aliquots were washed twice in 1 mL of cold phosphate-buffered saline (PBS) (137 mM NaCl, 2.7 mM KCl, 1.5 mM KH_2_PO_4_, 4.3 mM Na_2_PO_4_, pH 7.4) and centrifuged at 9000× *g* for 30 s. The pellet was resuspended in Laemmli sample buffer [31], boiled for 5 min, and centrifuged once more. Supernatants were then supplemented with 5% β-mercaptoethanol and boiled again for 5 min. Isolated proteins were separated on SDS-PAGE (60 mA, 120 min). Afterwards, proteins were transferred to polyvinylidene fluoride (PVDF) membranes at 250 mA for 60 min at 4 °C. Transfer was monitored by Ponceau red stain. The membranes were blocked with 5% milk in TRIS-buffer saline with 0.1% Tween 20 (T-TBS) and then incubated with the primary antibodies anti phospho-PRKA substrates (pPRKAs) (Cell Signaling Technologies, Danvers, MA, USA. #9624); total PRKAC (Cell Signaling Technologies, #4782); phospho-PRKAC (Thr197) (Cell Signaling Technologies, #4781); PRKAR1 (Cell Signaling Technologies, #3927); PRKAR2 (Epitomics, Burlingame, CA, USA. #1528-1); and AKAP3 (Santa Cruz Biotechnology, Dallas, TX, USA. #SC-135146). Membranes were incubated overnight at 4 °C with gentle shaking. After this step, membranes were washed six times and incubated with the appropriate biotinylated secondary antibody for 1 h at room temperature. Next, the membranes were washed with T-TBS for the last time, and blots were visualized by chemiluminescence (Amersham Corp., Sydney, Australia) according to the manufacturer’s instructions. Finally, the signal was imaged using an In-Vivo F Pro molecular imaging system (Bruker Corporation, Billerica, MA, USA). 

### 2.7. Stripping the PVDF Membranes 

To confirm equal protein load, blots were stripped and re-probed with an antibody against β-tubulin (Developmental Studies Hybridoma Bank, Iowa City, IA, USA. #AB-2315513). For this procedure, 15 mL of stripping buffer, consisting of 2% (*w/v*) SDS, 62.5 mM Tris, pH 6.7, and 100 mM 2-mercaptoethanol, was added to the membrane for 1 h with constant shaking at 60 °C. The membrane was then washed 6 times for 10 min in TBS, blocked, and probed with the primary antibody, as described above.

### 2.8. Immunofluorescence Assessment of Spermatozoa 

Spermatozoa were fixed in 4% paraformaldehyde for 15 min and washed twice by centrifugation with 0.1 M glycine in PBS at 14,000× *g* for 2 min. Then, the cells were permeabilized for 10 min with 0.1% Triton X-100 diluted in PBS and washed twice with PBS at 14,000× *g* for 2 min. The final pellet was resuspended in PBS supplemented with 1% BSA for 30 min. The sperm suspensions were spread on a glass slide and were allowed to dry for 15 min at room temperature. Then, the samples were incubated overnight with the primary antibodies for pPRKAs (dilution 1:50), PRKAR1 (dilution 1:20), PRKAR2 (dilution 1:100), and AKAP3 (dilution 1:20). Samples were washed 3 times with PBS and incubated with chicken anti-rabbit conjugated to Alexa 594 antibody (Thermo Fisher Scientific, Waltham, MA, USA. #A-21442) or goat anti-mouse conjugated to Alexa 488 antibody (Thermo Fisher Scientific, #A-10680) diluted 1:500 in 1% BSA-PBS for 2 h at room temperature. Finally, the samples were washed 3 times for 5 min with PBS, and the coverslips were mounted with Dako Fluorescent Mounting Medium and examined under a confocal microscope (Leica TCS SP8 SMD, Mannheim, Germany). A minimum of 400 cells were analyzed in each sample. Controls using a secondary antibody alone were performed to assure specificity.

### 2.9. Statistical Analyses 

Data were assessed by the Kolmogorov–Smirnov test and were normally distributed. They were analyzed by a one-way analysis of variance with Tukey’s post hoc test. Densitometry analysis was performed using the image 2.0j software and normalized with regard to the internal control tubulin. 

Quantitative results are presented as mean values and SEM. In all cases, results with *p* < 0.05 were considered as values with significant differences.

## 3. Results

### 3.1. The Proteasome Is Required for PRKA Signaling during Human Sperm Capacitation

The activation of PRKA during capacitation has been widely studied, and it has been defined as a fundamental event for sperm to acquire its fertilizing capacity. In this study, PRKA activity was evidenced by the phosphorylation of its catalytic subunit Thr197 [32,33,34] (Figure 1A) and specific presence of pPRKAs by Western blot (Figure 2). The results show that the intensity of the phospho PRKAC band of sperm incubated in NCM (0 min) was lower than those incubated in CM (Figure 1A, *p* < 0.01). In sperm incubated for 1 min in CM, a rapid increase in the intensity of the phospho PRKAC band was observed. The intensity of this band remained high and constant for the next 60 min. In the presence of the proteasome inhibitor epoxomicin, the intensity of the phospho PRKAC band was unaffected for the first 15 min. However, sperm incubated for 30 and 60 min in the presence of epoxomicin exhibited a significant decrease in the intensity of the phospho PRKAC band (*p* < 0.01) compared to the controls without inhibitor. As expected, when sperm were incubated for 60 min in CM in the presence of the PRKA inhibitor, H89, a significant decrease in the intensity of the phospho PRKAC band was observed (Figure 1).

In the next experiments, PRKA activity and its regulation by the proteasome was evidenced by the appearance of pPRKAs and the effect of epoxomicin, respectively. Sperm incubated in NCM showed low levels of pPRKAs (Figure 2A, T0 min). Sperm cells incubated in CM showed high levels of pPRKAs, starting at 1 min and remaining high for the next 60 min. In contrast, sperm incubated for 30 and 60 min in the presence of epoxomicin showed a significant decrease in the intensity of the pPRKAs bands (Figure 2A, *p* < 0.001). 

The next experiment was designed to determine the effect of epoxomicin on the subcellular localization of pPRKAs in human sperm (Figure 3). Sperm incubated in NCM (T0) presented pPRKAs labeled exclusively in the flagellum (Figure 3B with less intensity and a lower percentage than in sperm incubated in CM for 60 min (T60, Figure 3F) (65 ± 9% vs. 91 ± 2, respectively. *p* < 0.01). Although most of the sperm incubated in CM for 60 min presented labeling in the flagellum, only 24 ± 6% had labeling in the equatorial segment of the head (Figure 3F). Sperm incubated for 60 min in the presence of epoxomicin (T60 + Epox, Figure 3J), presented labeling on the flagellum (86 ± 8%) similar to that observed in the control without inhibitor and with the same fluorescence intensity. However, these sperm did not show labeling in the equatorial segment of the head (Figure 3J).

### 3.2. AKAP3 Is Degraded during Sperm Capacitation 

The following experiments were designed to study the possible molecule(s) that could be degraded by the proteasome and that regulate PRKA activity during capacitation. One likely candidate is AKAP3, which was shown to be degraded by the proteasome during bovine sperm capacitation [35]. First, we evaluated the localization of AKAP3 in human sperm (Figure 4A–D). The results reveal that most of the spermatozoa (91 ± 4%) exhibit labeling in the flagellum with a strong mark in the midpiece (Figure 4D). Furthermore, AKAP3 was observed in the sperm head with a strong label in the acrosomal region and in the equatorial line (96 ± 3%).

Next, we performed a capacitation kinetic of the protein levels of AKAP3. AKAP3 protein levels remained constant for 60 min of incubation in CM (Figure 5A). Subsequently, at 180 and 300 min of incubation, a significant decrease in AKAP3 protein levels was observed (Figure 5B; *p* < 0.01). At the end of the incubations (180 and 300 min), no significant decrease in the motility or viability of the sperm was detected (data not shown), indicating that the decrease in AKAP3 levels was not due to sperm death. Moreover, the protein levels of AKAP3 were evaluated during capacitation in the presence of 10 µM epoxomicin (Figure 5A, last right lane). Those sperm that were incubated for 300 min in the presence of epoxomicin did not show a decrease in AKAP3 protein levels (Figure 5B; *p* < 0.01). These results strongly suggest that the decrease in AKAP3 protein levels was due to proteasome degradation during human sperm capacitation. However, the degradation of AKAP3 started at 180 min, and, taking into account that the effect of the proteasome on PRKA activity was observed after 30 min, this degradation would not explain the regulation of the PRKA activity by the proteasome.

### 3.3. PRKAR1 Is Degraded during Sperm Capacitation 

PRKA is composed of two catalytic subunits (PRKAC) associated with two regulatory subunits (PRKAR1 and PRKAR2). Considering that the catalytic subunits of PRKA remained inhibited by the interaction with the regulatory subunits, the next step was to evaluate whether these regulatory subunits were degraded by the proteasome. The results showed that a molecular weight band of 52 kDa was observed at the different capacitation times that correspond to the PRKAR2 protein. This band did not show significant variations during 60 min incubation in CM (Figure 6A). Regarding its location, PRKAR2 is located only in the flagellum (88 ± 8%) with a strong intensity in the middle piece (Figure 6D). None of the analyzed sperm had a mark on the head.

Next, we evaluated the protein levels of PRKAR1. A 48 kDa molecular weight band corresponding to PRKAR1 was observed (Figure 7A). The results showed that PRKAR1 did not present significant variations in their protein levels up to 15 min of incubation in a capacitating medium (Figure 7B). As can be seen, protein levels of PRKAR1 began to decrease significantly after 30 min of capacitation (Figure 7B; *p* < 0.01). However, in three out of nine experiments with different donors, PRKAR1 protein levels began to decrease after 60 min (data not shown). At the end of the incubation, we could not detect any significant decrease in sperm motility and viability, demonstrating that the decrease in PRKAR1 levels was not due to cell death. We then investigated whether the decrease in PRKARI levels was mediated by proteasomal activity. In fact, the decrease in PRKAR1 after 60 min of incubation in capacitating conditions was significantly inhibited by the presence of epoxomicin (Figure 7A, last right lane; Figure 7B; *p* < 0.01), indicating that the degradation of PRKAR1 occurred through the proteasomal machinery.

When evaluating the location of PRKAR1 (Figure 7C–F), it was observed that this protein was located both in the flagellum and in the head of human sperm (92 ± 5%), which is consistent with proteasomal location in these cells (Figure 7F). 

## 4. Discussion

Proteasomes play important roles during sperm capacitation [16,17,18]. In the present study, we show that the proteasome degrades PRKAR1 in human sperm. This is based on the finding that the proteasomal inhibitor epoxomicin significantly inhibited the decrease in PRKAR1 levels. Degradation of PRKAR1 by the proteasome could explain the maintenance of PRKA activity in spite of low intracellular levels of cAMP during sperm capacitation. It is well accepted that the reproductive success of mammalian sperm requires precise orchestration of multiple events, all of them regulated through the cAMP-dependent PRKA pathway. PRKA activation is a necessary element for sperm capacitation; sperm from animals lacking the PRKAC subunit are infertile, despite normal mating behavior [23]. In the present study, we report that PRKA activity increased rapidly one minute after the start of capacitation and then remained high until 60 min of capacitation compared to spermatozoa incubated in non-capacitating conditions. These results are consistent with those in prior published work [26,27]. The activation of PRKA was followed by an increase in pPRKAs. The results obtained with the antibody that recognizes pPRKAs showed an immediate increase after 1 min of sperm capacitation, similar to that described by other authors [28,36]. The increase in PRKA activity reached its peak after 5 min and remained constant up to 60 min. This constant phosphorylation observed up to 60 min of capacitation is consistent with that described by Battistone et al. [28]. The phosphorylated bands detected with the anti pPRKAs antibody during capacitation were dependent on PRKA activity, since in the presence of a PRKA inhibitor the bands did not increase and there were no differences with spermatozoa incubated in NCM. Regarding the subcellular localization of pPRKAs in human sperm, pPRKAs were detected at the flagellar level in non-capacitated and capacitated sperm. However, a higher intensity of fluorescence was obtained at the flagellar level in capacitated spermatozoa. Several studies have shown that during capacitation, flagellar proteins are phosphorylated in tyrosine residues and that this step is necessary for the acquisition of hyperactive motility [37]. Considering that in sperm, PRKA phosphorylates various proteins in Ser and Thr residues to finally activate a signaling cascade leading to increased phosphorylation of proteins in tyrosine residues [38], it is correct to assume that this Tyr phosphorylation depends on PRKA activity. Battistone et al. [28] described that pPRKAs localize only at the flagellar level with no variation during capacitation. However, our results indicate that 24 ± 6% of the capacitated sperm have a mark in the equatorial segment of the head. This localization of pPRKAs is consistent with the localization of PRKA [16,39,40,41], AKAPs [42], and with phosphorylation in Ser and Thr residues, described at the level of the head by Naz [43] in human spermatozoa.

In our study, the inhibition of proteasome activity early during capacitation decreased phosphorylation and changed the subcellular localization of pPRKAs in human sperm. Something similar has been described in later events of capacitation. Human sperm incubated for 18 h in the presence of proteasome inhibitors exhibited changes in the protein phosphorylation pattern at Ser and Thr residues [44]. When evaluating the activity of PRKA, it increased significantly after 1 min of capacitation, remaining high throughout the incubation. However, epoxomicin treatment significantly decreased PRKA activity after 30 min of capacitation. These results indicate that the proteasome regulates the maintenance of PRKA activity during early capacitation. As a precedent, PRKA activity is regulated by the proteasome in other cellular systems. In Aplysia, the proteasome has been shown to regulate PRKA activity during synaptic plasticity [45]. This has also been shown in neuroblastoma cells, where the proteasome regulates the maintenance of PRKA activity during long-term memory [46]. Our group recently published that during the early capacitation of human sperm, the activity of the proteasome increases after 5 min of capacitation and is regulated by the SACY/cAMP/PRKA pathway [16]. In this new study, we present evidence that the maintenance of PRKA activity is regulated by the proteasome after 30 min of capacitation, which leads to the regulation of pPRKAs. Given this background, we will now discuss the possible mechanism of PRKA activity regulation by the proteasome during early events of capacitation.

The activation of PRKA depends on its adequate localization in subcellular regions. AKAP proteins are responsible for targeting PRKA and other proteins to specific subcellular locations [47]. AKAPs play important roles in sperm function, including capacitation and acrosome reaction [48]. Hillman et al. [35] demonstrated that AKAP3 is degraded by the proteasome after 4 h of capacitation in bovine sperm. Furthermore, Vizel et al. [48] demonstrated that this degradation is important for capacitation to occur in bovine sperm. Here, we present evidence that AKAP3 is degraded by the proteasome during human sperm capacitation. However, the degradation of AKAP3 occurs after 3 h of capacitation. Thus, this degradation does not explain the maintenance of PRKA activity by the proteasome during capacitation, an effect that was observed after 30 min. This degradation of AKAP3 may be related to the acrosomal reaction that corresponds to a post-capacitation event. In human sperm, it has been shown that inhibition of the proteasome decreases the percentage of sperm that undergo the acrosomal reaction [44] and that PRKA plays a role during this event [49].

PRKA is a tetrameric enzyme consisting of two catalytic subunits (PRKAC), which are maintained in an inactive state by binding to regulatory subunits (PRKAR). In mice, the elimination of PRKAR2 improves the basal activity of PRKAC, leading to an increase in the metabolic rate in adipose tissue [50]. In other cellular models, regulatory subunits have been shown to be degraded by the proteasome. In Aplysia sensory neurons, PRKAR is degraded by the proteasome, with consequent persistent activation of PRKAC during synaptic plasticity [45]. Lignitto et al. [46], demonstrated that the PRKAR subunit is ubiquitinated, which allows the proteasomal degradation of this subunit. The ubiquitination of PRKAR is through a signaling mechanism involving cAMP/PRKA. Furthermore, they demonstrated that this degradation is necessary to maintain high PRKAC activity in long-term memory processes [46] through a positive feedback mechanism between PRKA and the proteasome.

In this study, we report by immunofluorescence analysis that PRKAR2 is located exclusively at the flagellar level, while PRKAR1 is located both at the flagellar level and in the head. These results are similar to those described in sperm from other species [51,52]. This differential location could be related to different roles for the PRKAR1 and PRKAR2 isoforms in sperm function. When evaluating the protein levels of these subunits, we observed that the PRKAR2 remained constant during incubation, suggesting that this subunit is not degraded in the early stages of capacitation. This correlates with the subcellular location of PRKAR2, since the proteasome in human sperm is not located at the flagellar level [14,16,53,54]. On the other hand, protein levels of PRKAR1 in human sperm decreased after capacitation for 30 min. This decrease in PRKAR1 levels was most probably due to the degradation of this subunit by the proteasome, since this process was inhibited by epoxomicin. In conclusion, our results show that the maintenance of PRKA activity is regulated by the proteasome after 30 min of capacitation, which is correlated with the degradation time of PRKAR1. These results indicate that PRKAR1 is a possible target of proteasome regulation of the maintenance of PRKA activity during early human sperm capacitation. Additional studies are needed to fully understand the relationship between PRKAR1 and the proteasome in mammalian sperm.

## Figures and Tables

**Figure 1 cells-10-03501-f001:**
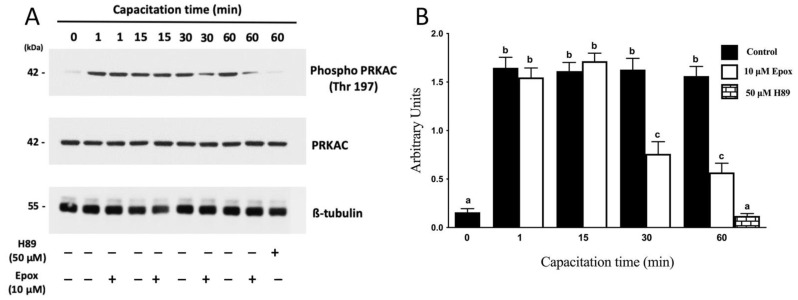
Effect of proteasome inhibition on the phosphorylation of PRKAC in Thr197 during human sperm capacitation. Human sperm were incubated for different times with 0.1% DMSO, 10 µM epoxomicin, or with 50 μM H89. Time 0 (0 min) corresponds to sperm incubated in non-capacitating conditions. (**A**) Western blot of PRKA activity evaluated using the anti-phospho Thr197 antibody against the catalytic subunit of PRKA (PKA-C). (**B**) For the densitometric analysis, total PRKAC/ß-tubulin levels were used as loading control. Bars represent the mean ± S.E.M of five different experiments. Different letters indicate statistically significant differences (*p* < 0.01) between groups.

**Figure 2 cells-10-03501-f002:**
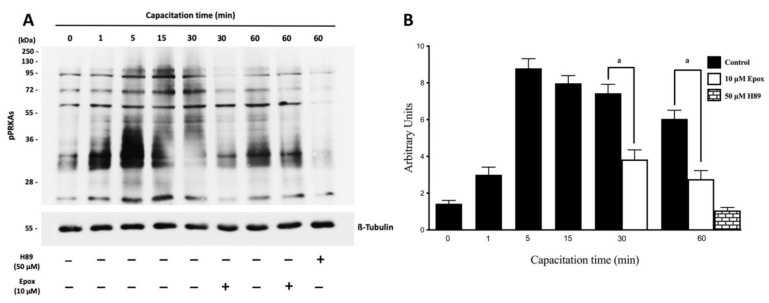
Effect of proteasome inhibition on the phosphorylation of PRKA substrates (pPRKAs) during human sperm capacitation. Human sperm were incubated for different times with 0.1% DMSO, 10 µM epoxomicin, or with 50 μM H89. The time 0 (0 min) corresponds to sperm incubated in non-capacitating conditions. (**A**) Phosphorylated PRKA substrates (pPRKAs) were evaluated by Western blot. (**B**) For the densitometric analysis, β-tubulin was used as loading control. Bars represent mean ± S.E.M. of seven different experiments. a indicates *p* < 0.001 vs. corresponding control.

**Figure 3 cells-10-03501-f003:**
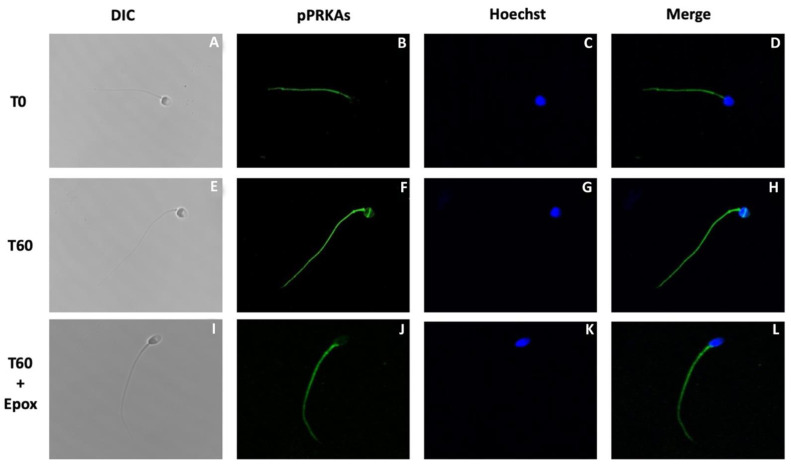
Effect of proteasome inhibition on the localization of phosphorylated PRKA substrates (pPRKAs) during sperm capacitation. Human sperm were incubated for 0 (T0) and 60 (T60) min with 0.1% DMSO or for 60 min with 10 µM epoxomicin (T60 + Epox). Cells were then fixed and labeled with a primary anti-PRKA phosphosubstrates antibody (**B**,**F**,**J**; green) and with Hoechst (**C**,**G**,**K**; blue). DIC: differential interference contrast (**A**,**E**,**I**). Merge: merged image of phosphorylated PRPKA substrates and Hoechst (**D**,**H**,**L**; green and blue).

**Figure 4 cells-10-03501-f004:**
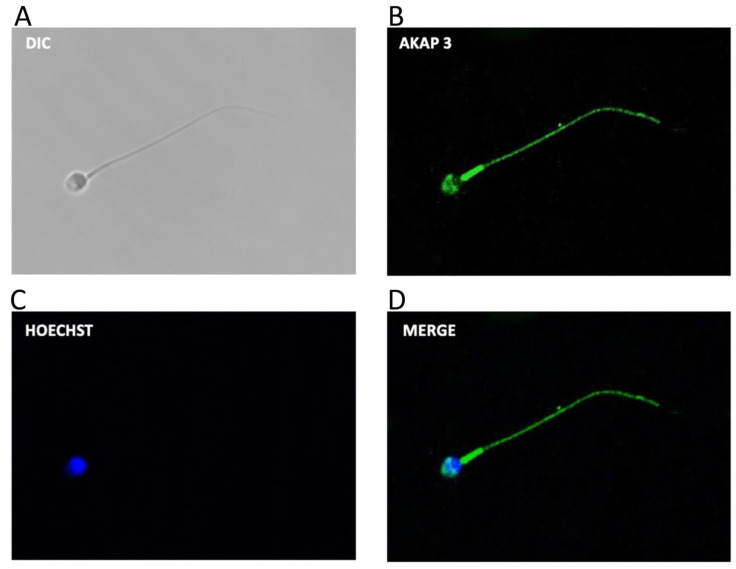
Subcellular location of AKAP3 during human sperm capacitation. Sperm suspensions were incubated under capacitating conditions for 60 min. Then, the cells were fixed and labeled with a primary anti-AKAP3 antibody (**B**, green) and with Hoechst (**C**), (blue). DIC: differential interference contrast (**A**). Merge: merged image of AKAP3 and Hoechst ((**D**); green and blue).

**Figure 5 cells-10-03501-f005:**
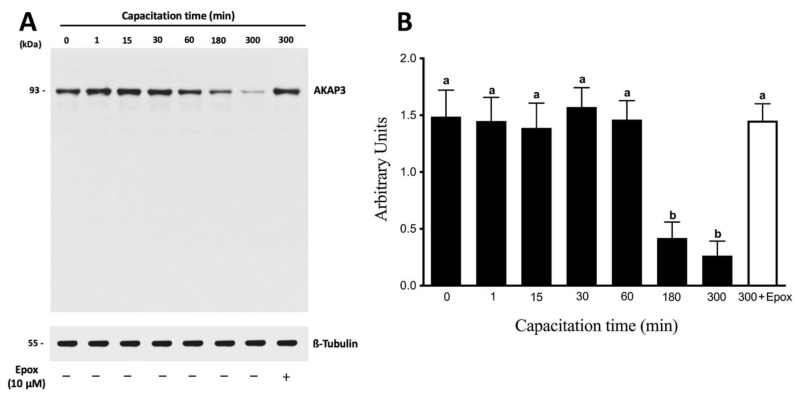
The degradation of AKAP3 is dependent on the sperm proteasome. Total protein extracts were obtained from sperm incubated at different times with 0.1% DMSO or for 300 min with 10 μM epoxomicin (Epox). (**A**) The level of AKAP3 was detected by Western blot using an anti-AKAP3 antibody. (**B**) For the densitometric analysis, β-tubulin was used as a loading control. Bars represent the mean ± S.E.M of five different experiments. Different letters indicate statistically significant differences (*p* < 0.01) between the groups.

**Figure 6 cells-10-03501-f006:**
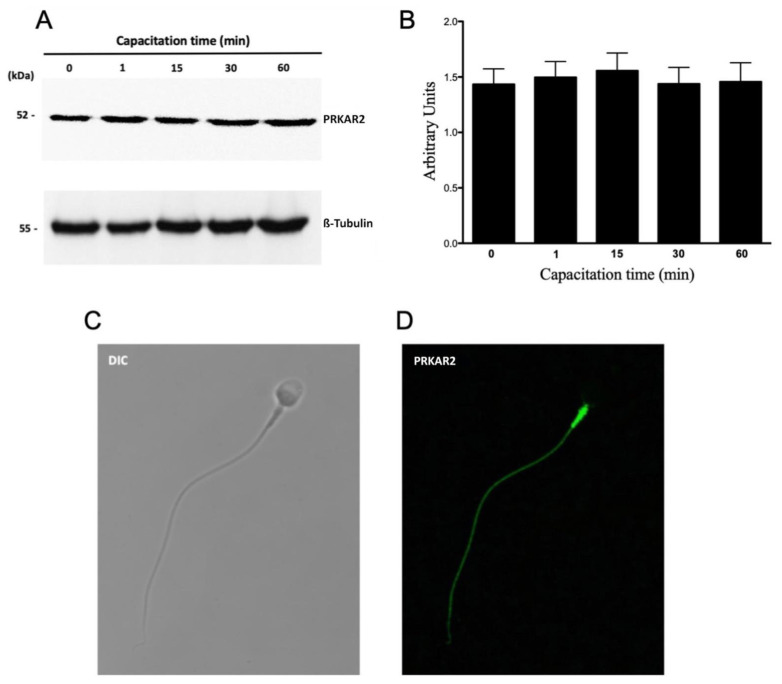
Protein levels and subcellular localization of PRKAR2 during human sperm capacitation. (**A**) Total protein extracts were obtained from sperm incubated for 0, 1, 15, 30, or 60 min. The level of PRKAR2 was detected by Western blot using an anti-PRKAR2 antibody. (**B**) For the densitometric analysis, β-tubulin was used as a loading control. Bars represent the mean ± S.E.M of six different experiments. (**C**,**D**) For subcellular localization, the sperm were incubated under capacitated conditions for 1 min. Then, the cells were fixed and labeled with a primary anti-PRKAR2 antibody (**D**, green). DIC: differential interference contrast (**C**).

**Figure 7 cells-10-03501-f007:**
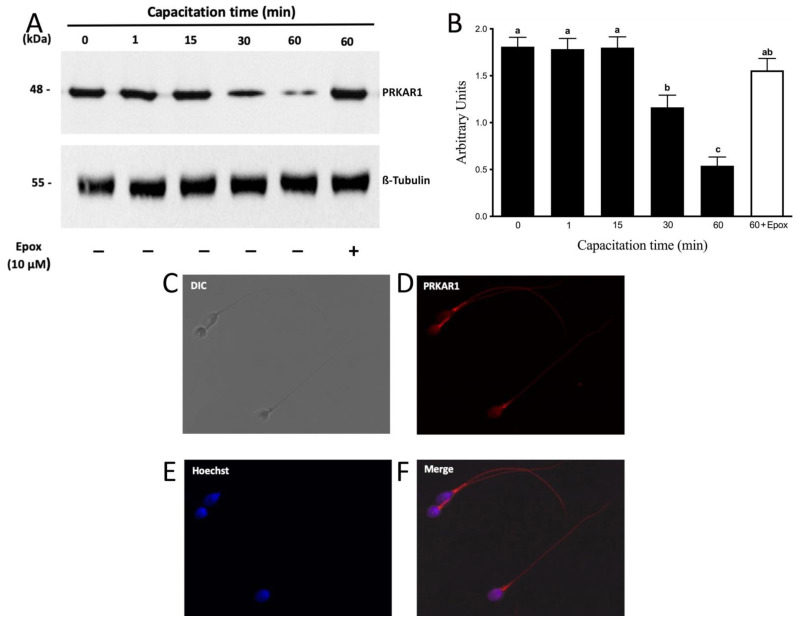
The degradation of PRKAR1 is dependent on the sperm proteasome. Total protein extracts were obtained from sperm incubated at different times with 0.1% DMSO or for 60 min with 10 µM epoxomicin (Epox). (**A**) The level of PRKAR1 was detected by Western blot using an anti-PRKAR1 antibody. (**B**) For the densitometric analysis, β-tubulin was used as a loading control. Bars represent the mean ± S.E.M of seven different experiments. Different letters indicate statistically significant differences (*p* < 0.01) between groups. (**C**–**F**) For subcellular localization of PRKAR1, sperm were incubated under capacitation conditions for 1 min. Then, the cells were fixed and labeled with a primary anti-PRKARI antibody (**D**, red) and with Hoechst (**E**, blue). DIC: differential interference contrast (**A**). Merge: merged image of PRKAR1 and Hoechst (**F**; red and blue).

## Data Availability

Not applicable.

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
