# Peer review of "Protein Kinase A (PRKA) Activity Is Regulated by the Proteasome at the Onset of Human Sperm Capacitation"

_cells, 2021, doi:10.3390/cells10123501_

Round 1

Reviewer 2 Report

In the MS entitled “Protein Kinase A (PKA) Activity is Regulated By the Proteasome at the Onset Of Human Sperm Capacitation” by Zapata-Carmona et al., the authors have conducted interesting research that shows the target of PKA regulation by the proteasome in human sperm. It is worth mentioning that these authors have been devoted to investigations into the role of the sperm proteasome in regulating PKA activity once human sperm capacitation has started.

The paper is very interesting and falls into the scope of Cells, it is novel and merits publication. I have no major comments about the experimental design, length, and content of the manuscript, but only specific comments that hope will help the authors to revise their manuscript. It should be noted that the Discussion is well written and covers all the results obtained in the study. My recommendation is for publication after Minor Revision.

Please correct or explain the following lines:

Introduction

  • L77: “treatment with inhibitors” I understand that you are talking about your work so if you only use one proteasome inhibitor (epoxomicin) you shouldn’t use the plural form.

Materials and methods:

  • Semen collection:

N=27…but

¿How many ejaculates were evaluated in your work?

¿How many donors were used in your work?

How many donors were used twice or more times in your work?

Did you exclude any samples based on any criteria?

When we work with human sperm samples it is very important to know these dates to calculate individual variations.

L147- You should show the sperm viability results throughout the experiment.

Results:

  • What other mechanisms could be influencing PKA phosphorylation is not reduced more drastically

They must justify which are the other pathways by which this protein can be phosphorylated and which are not influenced by the action of the proteasome.

  • The densitometry graphs are not very clear.

Do black bars represent tubulin control or PKA total protein control?

Do the white ones represent PKC activity?  Indicate this in a legend for all corresponding figures.

-Regarding AKAP3 degradation:

  • AKAP3 degradation does not occur at the same time as PKA activity, so perhaps there are other proteins involved in the process. Have you tried other pathways related to the phosphorylation of PKA during the so-called sperm capacitation? For instance, AMPK protein (Calle-Guisado, V., de Llera, A. H., Martin-Hidalgo, D., Mijares, J., Gil, M. C., Alvarez, I. S., Bragado, M. J., & Garcia-Marin, L. J. (2017). AMP-activated kinase in human spermatozoa: identification, intracellular localization, and key function in the regulation of sperm motility. Asian journal of andrology, 19(6), 707–714. https://doi.org/10.4103/1008-682X.185848)

  • Regarding capacitation process:
  • The sperm capacitation is manifested by certain patterns of motility, these have not been shown. Have you observed the change in this pattern of mobility in the times in which you are marking that this capacitation happens? Do you have those motility graphs with the parameters that capacitation is defined (hyperactivated motility, changes in WOB, and DANCE…)?

Regarding degradation of PKA RII:

  • According to your results, PKA RII degrades during training, but the level in WB remains the same during all measured times.

I have not understood this point well.

Could you explain to me or explain at work what is the behavior of this regulatory subunit. Why do you include it in the results if it doesn't seem to be mediated by the proteasome?

Thanks and congrats.

Round 2

Reviewer 1 Report

Dear authors,

after reading the revised version of your manuscript, I believe that the changes made answer to the remarks I proposed. 

I noticed typos that I raise to your attention so you can correct them during the proofreading process :

  • Line 223 in the legend of Figure 1 the term "analisis" remained in spanish ;
  • Line 375 the comma placed between "flagellar, level" seems useless.

Sincerely.

Author Response

We have changed the manuscript according to the reviewer suggestions.